# Novel Aspects Targeting Platelets in Atherosclerotic Cardiovascular Disease—A Translational Perspective

**DOI:** 10.3390/ijms24076280

**Published:** 2023-03-27

**Authors:** Aydin Huseynov, Julius Reinhardt, Leonard Chandra, Daniel Dürschmied, Harald F. Langer

**Affiliations:** Department of Medicine, Cardiology, Angiology, Hemostasis and Intensive Care Medicine, University Medical Center Mannheim, Faculty of Medicine Mannheim, University of Heidelberg, 68135 Mannheim, Germany

**Keywords:** platelets, atherosclerosis, coronary artery disease, myocardial infarction, neuroinflammation, ischemic stroke, neurovascular disease, thrombosis

## Abstract

Platelets are important cellular targets in cardiovascular disease. Based on insights from basic science, translational approaches and clinical studies, a distinguished anti-platelet drug treatment regimen for cardiovascular patients could be established. Furthermore, platelets are increasingly considered as cells mediating effects “beyond thrombosis”, including vascular inflammation, tissue remodeling and healing of vascular and tissue lesions. This review has its focus on the functions and interactions of platelets with potential translational and clinical relevance. The role of platelets for the development of atherosclerosis and therapeutic modalities for primary and secondary prevention of atherosclerotic disease are addressed. Furthermore, novel therapeutic options for inhibiting platelet function and the use of platelets in regenerative medicine are considered.

## 1. Introduction

Atherosclerosis still is the major cause of death in developed countries and accounts for an estimated 17.9 million deaths every year [1,2] During atherogenesis, the initiation of immune processes, a sustained inflammatory reaction and platelet activation with subsequent intravascular thrombus formation are major contributors to the pathogenesis of this detrimental disease [3] Furthermore, the development of atherosclerotic lesions requires elevated low-density lipoprotein cholesterol, an elevation in triglyceride-rich lipoproteins and low high-density lipoprotein [4]

Risk factors for atherosclerosis and its thrombotic complications include hypertension, cigarette smoking and diabetes mellitus [3,5] Besides the classical risk factors, research of the last decades indicates that inflammation is an additional key aspect of atherogenesis and of the pathophysiology causing cardiovascular disease [6] This involves the complex interaction of diverse cell types (e.g., macrophages, monocytes, lymphocytes, thrombocytes) and inflammatory mediators, coagulation proteins as well as cytokines building signalling pathways that trigger and sustain the atherosclerotic process [7] Here, the role of thrombocytes appears to be one of the less obvious roles in the activation chain. Nevertheless, it becomes more and more evident that platelets not only contribute to thrombus formation through activation of coagulation pathways, but they also release potent proinflammatory mediators from their granules [8].

Treatment concepts for atherosclerosis and associated cardiovascular diseases are diverse and have one goal of interrupting the parts of the chain that lead to the development of atherosclerosis. From the promotion of a healthy lifestyle to reduce the influence of cardiovascular risk factors to prophylactic medical treatment in risk patient groups, modern concepts are available for the prevention and treatment of arteriosclerosis [9,10] In addition to drug therapy, there are also interventional and surgical methods to treat the consequences of atherosclerosis. Typically, a combination of different therapeutic approaches is used in most patients to treat existing atherosclerotic plaques and prevent further development of this chronic disease.

This review focuses on recent translational aspects concerning the role of platelets for cardiovascular disease, and at the same time emphasizes the practical aspects of anti-platelet strategies as diverse treatment options for acute and chronic atherosclerotic disease.

## 2. Atherosclerosis

During atherogenesis, an atheromatous plaque develops in the inner lining of the arteries. The modern concept of atherogenesis is based on initial qualitative changes within the layer of endothelial cells lining the inner arterial surface. Because of various risk factors, such as dyslipidemia, hypertension or inflammatory conditions, the endothelial cells express important adhesion molecules on the surface for the recruitment of leukocytes and other cells [11] Parallel changes in endothelial permeability promote entry and retention of cholesterol-containing low-density lipoprotein (LDL) particles in the arterial wall [12] Biochemically modified components of these particles promote leukocyte adhesion. Endocytosis of such particles from macrophages, derived from monocytes, results in an intracellular accumulation of cholesterol (Figure 1) [13] These monocytes then differentiate into tissue macrophages and remain in the plaque [14]

According to current data, a central role in plaque formation is attributed to oxidized LDL [15] While the direct connection is mainly proven by in vitro and animal experiments, the clear causal proof data in humans have not been provided so far. In this context, several pathophysiological mechanisms are possibly involved eventually causing atherosclerotic disease, such as the activation of fibrinolysis in the context of plasminogen binding by oxidized lipids [16] Other mechanisms are based on immune responses to native LDL and LDL-receptor-mediated uptake of intimal cells by macrophages [17,18].

Endothelial activation can be understood as a defense reaction with the expression of chemokines, cytokines and adhesion molecules designed to interact with each other and other blood components such as leukocytes and platelets [19,20] It has been shown that most cardiovascular risk factors can activate these defense mechanisms with the aim of protecting the arterial wall from damage and initiating repair mechanisms [21,22].

Atherosclerotic lesions tend to occur locally in the arterial curvature or bifurcation where shear stress of blood flow is low, the so-called disturbed flow [23] These flow patterns act on vascular endothelial cells to activate several inflammatory pathways, increasing permeability, oxidative stress and expression of inflammatory receptors and cytokines that recruit leukocytes [24] This turbulent flow promotes changes in the extracellular matrix that contribute to endothelial inflammation. Inflammatory factors and other aspects of matrix remodeling increase the infiltration and retention of lipoproteins, known to trigger atherogenesis [25,26,27] Immune cells dominate early atherosclerotic lesions, their effects accelerate lesion progression, and activation of inflammation can lead to acute cardiovascular events [19] Activated platelets play a special role in triggering the inflammatory reactions [28] They induce primary hemostasis and clot formation after rupture of atherosclerotic plaques in acute cardiovascular events, such as stroke or myocardial infarction, and on the other hand they mediate and trigger pro-inflammatory pathways [29,30,31] Due to the complexity of atherogenesis, only some components of its pathogenesis have been identified as therapeutic targets. Among them is lipid-lowering therapy, and furthermore anti-inflammatory substances have found their way into clinical practice.

### 2.1. Platelet Activation, Adhesion, Thrombus Formation and Consecutive Development of Chronic and Acute Disease

In the steady state with an intact intima of the blood vessels, there is no contact between platelets and endothelial cells; however, in the context of vascular injury, platelets adhere to the subendothelial extracellular matrix to limit bleeding and promote tissue healing [32].

### 2.2. Platelet Activation and Adhesion

Adhesion is triggered by various proteins localized to the subendothelial matrix, the most important being collagen, von Willebrand factor (VWF), laminin, fibronectin and thrombospondin. Platelet adhesion occurs through interaction between these proteins and corresponding platelet receptors [33].

Various collagen types (I, III, IV and VI) can elicit platelet responses, interacting directly with two platelet membrane glycoproteins (GPs), integrin-α2β1 (GPIa–Iia) and GPVI (Figure 1 and Figure 2) [34] One of the most important components is the von Willebrand factor, which is responsible for the formation of the complex with platelets and the absence of the factor causes defects in primary hemostasis and coagulation [35].

Abbreviations: GP, glycoprotein; LMWH, low-molecular-weight heparin; PAR, protease-activated receptor; PDE, phosphodiesterase; TP, thromboxane prostanoid; TXA2, thromboxane A2; UFH, unfractionated heparin; VWF, von Willebrand factor; 5HT, 5-hydroxytryptamine.

After the initial adhesion of platelets to the exposed endothelial surface, various autocrine and paracrine mediators. such as adenosine diphosphate, thrombin, epinephrine, and thromboxane A2. are released. The recruitment of additional platelets into the thrombotic plaque requires these mediators as well as the plasmatic coagulation factor thrombin, which act through G-protein-coupled receptors [39,40] Finally, platelet integrin glycoprotein (GP) IIb/IIIa receptor is activated, which is considered to be a major receptor for adhesion and aggregation [41] The GP Ib/IX/V complex plays an essential role in the initial phase of the platelet–vessel wall interaction, leading to activation of the GP IIb/IIIa integrin and platelet adhesion. These signaling pathways that regulate GP Ib/IX/V may be potential targets for antithrombotic drug development.

Furthermore, GP IIb/IIIa leads to the binding of fibrinogen and von Willebrand factor, which then mediates platelet crosslinking and thrombus formation [42].

### 2.3. Platelet Aggregation and Thrombus Formation

As one product of the plasmatic coagulation cascade, fibrinogen increases the stability of a thrombus by binding platelets through glycoprotein IIb/IIIa integrin bridges, this creates a crosstalk between the primary hemostasis and the plasmatic coagulation system [39].

The next phase of platelet activity, called aggregation, is integrated with the final stages of the coagulation cascade. Aggregation is mainly mediated by GP IIb/IIIa receptors on the platelet surface [43] These receptors are upregulated during platelet activation and are adhesion molecules for fibrinogen. The more such receptors are activated, the more fibrinogen is bound. At the same time, the GP IIb/IIIa receptors immobilize other soluble adhesion proteins, such as VWF, fibronectin and vitronectin, on the surface of adherent platelets. These proteins and fibrinogen then strengthen the structure of the thrombus [44].

The central role for the inhibition of platelet-mediated thrombus formation is played by the healthy endothelium. The main inhibition pathways provided by endothelial cells include nitric oxide, prostacyclin and Ecto-ADPase/CD39/NTPDase pathways. Each of these factors is a specific mechanism by which platelet function is inhibited [45].

### 2.4. Platelets and Development of Chronic and Acute Disease

Atherosclerosis is a slowly progressing disease, but often can transform from a stable and clinically silent disease to a symptomatic life-threatening condition. The usual mechanism for this development is the rupture and erosion of the endothelial surface or plaque destruction followed by thrombosis [39] Plaques containing a soft atheromatous core are unstable and may rupture, whereby the highly thrombogenic plaque substances are suddenly exposed to the flowing blood. Such disrupted plaques are found among ≈75% of the thrombi responsible for acute coronary syndromes [46] Plaque destruction occurs most frequently, where the fibrous cap is thinnest and most infiltrated by foam cells [47].

Most of such plaque ruptures are resolved by local repair mechanisms and do not lead to thrombosis and consequent vessel occlusion. However, such episodes, as well as hemorrhage into the damaged plaque and platelet activation at the plaque surface, lead to the unpredictable and nonlinear progression of atherosclerosis [39,46] Atherothrombosis, which involves direct interaction between atherosclerotic plaque and arterial thrombosis, causes the majority of cardiovascular events with resulting clinical scenarios such as stable and unstable angina, acute myocardial infarction, ischemic stroke and peripheral arterial occlusive disease [48] Global data showed a prevalence of coronary artery disease (CAD) of approximately 5–8% and a prevalence of 10–20% peripheral artery disease (PAD), depending on study design, mean age, gender and geographic location. Together, CAD and PAD represent a significant medical and economic burden worldwide, which can be improved through better use of guidelines, modification of risk factors and drug or non-drug interventions [49].

## 3. Clinical Implications

### 3.1. Diagnosis

The early identification of atherosclerotic lesions at risk of deteriorating to myocardial infarction or other acute cardiovascular events remains challenging. Non-invasive imaging, such as ultrasound, computed tomography, magnetic resonance or molecular imaging techniques, can help to detect atherosclerosis better and earlier, monitor plaque growth and allow a verification of the efficacy of risk factor control [50].

Invasive angiography is the standard clinical imaging technique used to assess luminal stenosis qualitatively and quantitatively [51] The disadvantage of the method, apart from the invasiveness of the procedure, is the silhouette of the vessel wall against the lumen without direct evidence of intravascular and subendothelial atherosclerosis [52] “Upgrading” of the angiography with intravascular ultrasound (IVUS) or optical coherence tomography (OCT) allows additional information to be determined about the genesis of the narrowing [53] Both IVUS and OCT play key roles in interventional cardiology, but the use of both methods to identify a vulnerable plaque is limited by their invasive nature as well as the plaque imaging not always providing us with the information on plaque activity [50] In summary, non-invasive and, in some patients, invasive imaging can identify atherosclerotic plaques. Some techniques are suitable to identify unstable plaques and better estimate the risk of adverse events to choose the best treatment strategy.

### 3.2. Surgical/Interventional Treatment of Intravascular Thrombosis

Optimal medical treatment and revascularisation improve symptoms and increase survival in patients with coronary atherosclerosis [24] The indication for revascularization recommended by the guidelines in patients with stable coronary artery disease is the persistence of symptoms despite optimal drug therapy. Both interventional and surgical revascularization relieves angina and reduces the use of antianginal drugs and improves physical performance and quality of life compared to drug therapy alone [51].

The percutaneous coronary intervention (PCI) strategy compared to drug-only treatment in patients with stable coronary artery disease (SCAD), according to the current data, provides little or no benefit in terms of survival or myocardial infarction for an invasive strategy. However, one must consider the fact that up to 40% of the patients have switched from the drug therapy arm to the revascularization arm, which makes a clear comparison of the two strategies very difficult [51,54,55] The data situation remains heterogeneous, although the evidence for the superiority of an invasive strategy with modern stents of the latest generation is accumulating [56].

Another therapeutic option for the revascularization of coronaries affected by atherosclerosis is surgical revascularization. The superiority of surgical revascularization over drug therapy has confirmed a survival benefit, particularly in patients with SCAD and left main or three-vessel disease when the left anterior descending coronary artery (LAD) proximal coronary artery is involved [51,56].

Both revascularization methods have benefits and limitations. PCI is associated with lower procedural morbidity and mortality compared to coronary artery bypass graft surgery (CABG); however, this treatment method requires long-term antiplatelet therapy, which increases the risk of bleeding complications in certain patients. Neither interventional nor surgical revascularization protect against atherosclerotic lesions that may develop in the future. Surgical revascularization, however, is of an advantage when such future lesion do develop proximal to a bypass anastomosis [24] CABG provides more complete revascularization with survival benefit compared to PCI, especially in patients with extensive coronary atherosclerosis. The application of the left internal mammary artery to anastomosis LAD should be particularly mentioned, as it offers excellent long-term results [57,58].

Surgical revascularization and interventional therapy should under no circumstances be regarded as mutually exclusive therapy options, since both methods are also excellently used as a hybrid approach in the sense of combining the left internal mammary artery to the left anterior descending coronary artery and PCI to the remaining vessels [59,60] The best therapy strategy for complex coronary artery disease should be determined by a multidisciplinary decision-making team, composed of representatives from the fields of interventional and surgical treatment, who discuss the complex cases individually [51,61].

### 3.3. Anti-Platelet Therapy

The use of drugs that inhibit platelet function (Figure 2) is one of the main pillars of atherosclerosis therapy. The benefits and risks of such therapy are different depending on the indication and accompanying conditions.

### 3.4. Anti-Platelet Therapy in Individuals without Atherosclerotic Disease

The use of antiplatelet drugs to prevent cardiovascular events in individuals without established atherosclerotic disease should be considered for the primary prevention of cardiovascular events. However, the balance of the benefits and harms of platelet inhibition has been widely studied and is still controversial. In the primary prevention trials in patients with low cardiovascular risk, aspirin was associated with a reduction in adverse events and, at the same time, with significant increase in bleeding events [62,63,64] Consequently, the decision to use aspirin for primary prevention may need to be made on a balance of the patient’s risk of bleeding and benefit [65] In some patients with high cardiovascular risk and diabetes mellitus, but without any evidence of cardiovascular disease, the primary prevention is associated with a number needed to treat of 95 to prevent one major adverse ischemic event in 5 years [66] In summary, low-dose aspirin may be considered for primary prevention in the absence of clear contraindications for patients with diabetes mellitus at high cardiovascular risk [9] In contrast to aspirin, the data on the use of other antiplatelet substances for the purpose of primary prophylaxis are very sparse, and randomized studies on this topic do not currently exist.

### 3.5. Anti-Platelet Therapy in Individuals with Established Atherosclerotic Disease

In contrast to primary prevention, the benefits of aspirin therapy are overwhelming in patients with established CAD, such that aspirin is now a standard pill for a CAD patient. Aspirin therapy is associated with reductions in serious vascular events, including stroke and coronary events, and a 10% reduction in total mortality [9,67] A positive effect in the context of secondary prophylaxis was also demonstrated with another antithrombotic medication, namely with a P2Y12 inhibitor clopidogrel [68] Which of the two substances, aspirin or clopidogrel, is better suited for secondary prevention could be answered by meta-analysis of comparative studies. The all-cause mortality, vascular death and stroke do not differ, but the risk of myocardial infarction is marginally lower (the number needed to treat is 244) in patients receiving P2Y12 inhibitor compared with those receiving aspirin [69].

The other antiplatelet agent sarpogrelate (selective 5-HT_2_A receptor antagonist) has been used for secondary prevention in patients with peripheral artery disease in Japan, China and the Republic of Korea (South Korea) as an additional or alternative antiplatelet drug in patients with aspirin allergy or clopidogrel-associated high risk of bleeding [70] The use of a 5-HT_2_ receptor antagonist in other indications (stroke prevention, CAD prevention) is controversial, with evidence of non-inferiority to aspirin in stroke [71] or aggravation of myocardial ishemia and reperfusion injury in vivo [72].

The combination of aspirin and a P2Y12 inhibitor termed “dual antiplatelet therapy” (DAPT) is part of the prevention of thrombotic complications in patients with a variety of manifestations of CAD [73] A patient with CAD may have DAPT after myocardial revascularization (e.g., percutaneous coronary intervention (PCI) or surgical revascularization using coronary artery bypass surgery (CABG)) after acute coronary syndrome (ACS) or for secondary prophylaxis in a high-risk constellation [74,75] Each indication for DAPT therapy is associated with at least two challenges, namely the choice of the appropriate P2Y12 inhibitor and the duration of DAPT therapy with the aim of minimizing the risk of ischemic and bleeding complications. As the risk of bleeding correlates to DAPT duration, the current guidelines recommend the individual assessment of the advantages of a prolonged DAPT, depending on the cardiovascular history and the risk of bleeding under DAPT using risk scores [74,76].

The studies with stroke patients also come to similar conclusions; however, in contrast to CAD patients, the bleeding complications are significantly more serious with increased rates of intracranial bleedings and stroke hematoma enlargements [77].

To summarize, the combinations of various P2Y12 inhibitors and aspirin lead to a reduction in ischemic events, they increase the bleeding rates, so that this is only recommended for secondary prophylaxis in certain patient groups. Table 1 lists the most important studies with antiplatelet agents over the last 30 years.

## 4. Novel Approaches for Anti-Platelet Therapy

The disadvantages of conventional antithrombotic therapy such as a delayed onset and variability of its antiplatelet action are the reasons why other substances are being investigated. An optimal antithrombotic agent selectively inhibits platelet function at the site of atherosclerotic plaque injury (spontaneous or PCI-induced) without significantly affecting systemic hemostasis [74,107].

### 4.1. Glycoprotein VI

One of the novel platelet inhibitors is a competitive antagonist for collagen GPVI signaling called Revacept^®^ (Figure 2). It competes with endogenous platelet GPVI, binds to exposed collagen fibers and inhibits collagen-mediated platelet adhesion and aggregation selectively at the location of plaque rupture—without significantly affecting the function of circulating platelets [107,108] A recent randomized trial, however, showed that Revacept^®^ on top of standard treatment did not reduce myocardial injury in stable CAD undergoing PCI with a generally low rate of bleeding events in the treatment arms [107] Further studies are needed to investigate further treatment regimens of this hopeful approach to platelet inhibition.

### 4.2. Glycoprotein Ib/IX/V

Another target for platelet inhibition is the platelet receptor complex GPIb/IX/V, which induces vWF-associated thrombus formation. Caplacizumab (also known as ALX-0081) is a single-domain nanobody that inhibits the vWF-GPIb/IX/V interaction by blocking the vWF A1 domain [109] In Phase I studies in healthy subjects and stable angina patients undergoing percutaneous coronary intervention (PCI), Caplacizumab was well tolerated and effectively inhibited pharmacodynamic markers [110] Unfortunately, further studies did not prove the effectivity of Caplacizumab in patients with high-risk PCI, however this approach was successfully used in the treatment of thrombocytopenic purpura [111].

### 4.3. P2Y1 Receptors

In addition to P2Y12 receptors, platelets express P2Y1 receptors, which are responsible for platelet shape change [112] Both receptors P2Y12 and P2Y1 are necessary for complete platelet aggregation, so that selective P2Y1 inhibitors are also developed [113,114] BMS-884775 and MRS2500 are both selective P2Y1 platelet inhibitors and yielded promising results in preclinical and early clinical studies [109] The substances that inhibit both receptors P2Y1 and P2Y12 are also promising. Phase I and II studies with these substances are currently underway [115]

## 5. Targeting Thromboinflammation in Atherosclerosis

Anti-inflammatory therapy to prevent cardiovascular events can be effective in treating common conditions, such as atherosclerosis, post-infarction heart failure and stroke. So far, three IL-1-targeted agents have been approved: the IL-1 receptor antagonist anakinra, the soluble decoy receptor rilonacept and the neutralizing monoclonal anti-IL-1β antibody canakinumab [116] The Canakinumab Anti-inflammatory Thrombosis Outcomes Study (CANTOS) has shown that specific targeting of IL-1β can significantly reduce cardiovascular event rates without lipid or blood pressure lowering [117,118] In addition to cytokines, platelets are also possible targets for anti-inflammatory therapy. Platelets and neutrophils promote acute thromboinflammatory responses, forming neutrophil platelet aggregates contributing to microvascular obstruction. Some preclinical studies confirmed that targeting adhesion receptors on platelets inhibits neutrophil–platelet aggregates and improves microvascular dysfunction and inflammation [119] This interaction of neutrophils with platelets is mainly mediated through P-selectin and beta2 and beta3 integrins (CD11b/CD18, CD41/CD61) [120] Recent studies have confirmed that targeting P-selectin, GPIb, aIIbb3 as well as neutrophils (PSGL-1, Mac-1) inhibits neutrophil–platelet aggregates and improves microvascular dysfunction and inflammation. A phase 1 trial has confirmed the safety of inclacumab, a monoclonal antibody against P-selectin, and established that it does not extend bleeding time or impact platelet aggregation [119,121] Other sets of platelet receptors, such as C-type lectin-like-2 (CLEC-2) receptor, which binds podoplanin, the collagen receptor glycoprotein VI (GPVI) and soluble forms of the TREM-like transcript (TLT-1) receptor, are also future drug targets [122,123,124].

CD40L, also known as CD154, is a type II transmembrane protein belonging to the TNF (Tumor Necrosis Factor) superfamily. While CD40L is intracellularly localized in inactivated platelets, activated platelets express CD40L on their surface, allowing them to interact with CD40-expressing cells, such as other platelets or endothelial cells, in addition to lymphocytes and dendritic cells [125] Future studies will seek to identify medicinal products that block CD40L-mediated interactions of platelets with inflammation/immunity without affecting their hemostatic function [126].

## 6. Targeting Platelet and Endothelium Interactions

Endothelial cells regulate hemostasis by reducing platelet excitability through the production of nitric oxide (NO) and prostaglandin I2 (PGI2), which have a constant inhibitory effect on circulating platelets. NO directly stimulates guanylyl cyclase (GC) in platelets to induce production of cGMP, while PGI2 acts on IP receptors to stimulate adenylyl cyclase (AC) to produce cAMP [127].

Antiplatelet drugs, particularly cilostazol, an inhibitor of phosphodiesterase III (PDE III), are widely used for the treatment of ischemic stroke, transient ischemic attack and peripheral arterial disease (PAD) [128] Cilostazol induces improvement in endothelial function by increasing endothelial NO synthase (eNOS) activity and decreases NO inactivation by reducing oxidative stress [129] A positive effect on the endothelial cells has also been demonstrated with widely used drugs such as Angiotensin-converting enzyme inhibitors and statins [130] Losartan and valsartan caused a dose-dependent release of NO from platelets and endothelial cells, which then has an inhibitory effect on collagen-stimulated adhesion and aggregation of platelets [131] The anti-inflammatory effects of statins have been repeatedly demonstrated in clinical and experimental work, with a clear positive effect on endothelial cells [132,133] Another special feature of the interactions between platelets and endothelial cells is expressed by the blockade of platelet P2Y12 receptors, which also significantly increases the sensitivity of platelets to the inhibitory effects of both PGI2 and NO [127] Indeed, since PGI2 and NO have synergistic inhibitory effects on platelets, interaction with P2Y12 receptor blockers provides a potent triple synergistic effect: the NO, PGI2 and P2Y12 blockades are individually inhibitory, act synergistically together and as a trio provide potent platelet inhibition [134].

## 7. Novel Treatment Methods with Platelets

Clinical platelet research in recent decades has focused on investigating the substances that inhibit platelet function and developing optimal therapy regimens with inhibitory drugs. However, in recent years, evidence has accumulated that platelets and platelet properties, such as participation in inflammatory and regenerative processes, can also be applied as therapeutic approaches in regenerative medicine [135,136] The release of growth factors, cytokines and modulators of the extracellular matrix from platelets promote revascularization of the damaged tissue through induction of endothelial cells and neo-angiogenesis as well as the proliferation and activation of fibroblasts and mesenchymal stem cells [137,138].

The processed liquid fraction of autologous blood with a platelet concentration above the baseline is called platelet-rich plasma (PRP) [139] The use of PRP has become a novel treatment option in various aspects of medicine including orthopaedics, cardiothoracic surgery, plastic surgery, dermatology, dentistry, diabetic wound healing and even infertility [140] The treatment involves an injection of concentrated platelets locally to modulate tissue repair [141,142] The evidence of this treatment is different depending on the indication with the strongest evidence in orthopaedics, particularly in knee osteoarthritis and achilles tendinopathy [135,143] Further studies are needed to standardize the PRP preparation techniques and administration regimes in different indications.

If uncontrolled thrombus formation takes place in pathological conditions such as atherothrombosis or thrombosis within the microcirculation (found in atherosclerotic vessels) upon tissue injury, or as a consequence of adverse drug effects, the forming thrombus causes life-threatening occlusive cardio- and cerebrovascular pathologies. These diseases represent a major health burden throughout the world with increasing incidence. Major progress has been made by pharmacologically reducing platelet-dependent arterial and microcirculatory thrombosis, resulting in improved disease control and consecutively a reduction in morbidity and mortality. Despite these developments, a significant count of patients who are treated with antithrombotic and antiplatelet therapy suffer from thrombotic episodes with a high rate of re-occurrence in patients with a previous history of thrombosis. On the other hand, treatment with antiplatelet drugs and anticoagulants has to be carefully managed, as a considerable bleeding risk is inflicted as well, which ranges from smaller bleeding that reduces compliance to severe hemorrhage with increased morbidity and mortality. Thus, a major challenge for future approaches will be to further improve antithrombotic therapy based on improved agents and, importantly, disease-specific treatment regimens that target platelet-dependent thrombosis with a reduced risk of bleeding, thus balancing safety with efficacy. Together, despite intense research over the recent decades, platelets remain an important focus of basic research, translational and clinical studies. This is not at all surprising given the variety of platelet-related aspects in the clinical context of cardiovascular disease, from canonical platelet functions, interactions with other mediators and cells to the inhibition of platelet function and the use of platelets in the context of regenerative medicine. In particular, finding new targets to simultaneously improve the local therapeutic effect without acquiring an undesired systemic effect is a major challenge for future platelet research.

## Figures and Tables

**Figure 1 ijms-24-06280-f001:**
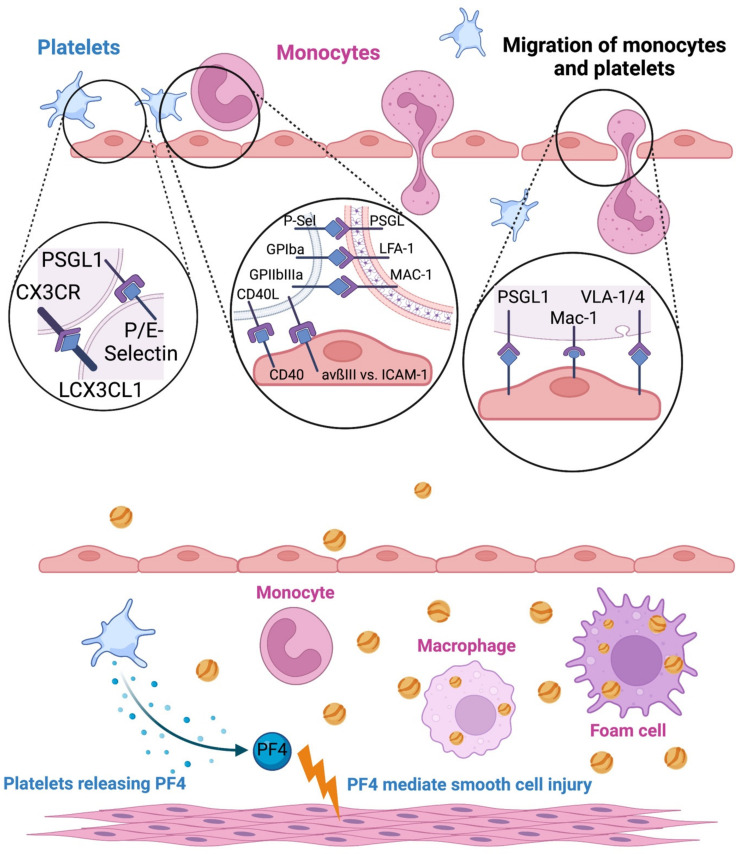
Platelet activation in atherosclerotic disease. Platelet endothelial and platelet monocyte interactions promote transendothelial migration. First, there is a binding between platelets and endothelial cells (CX3CR1 vs. CXCL1, PSGL1 vs. P-/E-selectin). Attached platelets bridge the connection of endothelial cells and monocytes (GPIba vs. LFA-1, GPIIbIIIa vs. MAX-1) and then both migrate between endothelial cells into the intima (mediated with endothelial P-Selectin, ICAM-1 and VCAM-1 receptors). In the intima, platelets release Platelet Factor 4 which recruit and mediate vascular smooth cells injury. Monocytes transform to macrophages and further to foam cells by picking up oxidized LDL particles.

**Figure 2 ijms-24-06280-f002:**
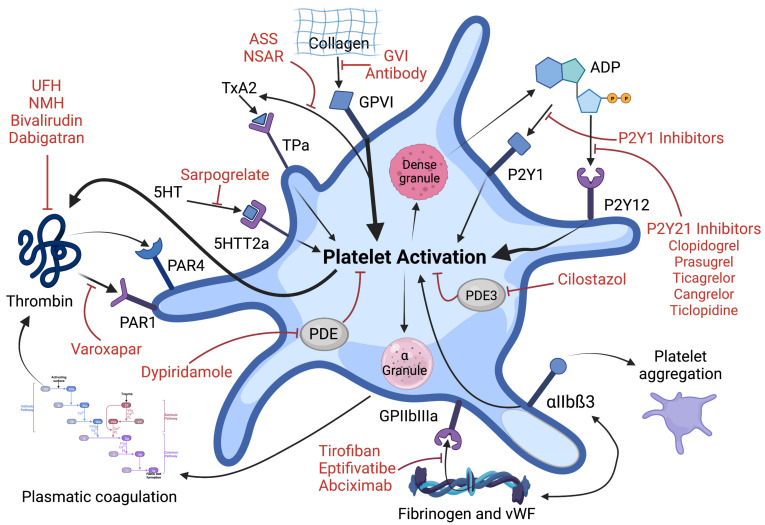
Antiplatelet strategies. Platelet activation pathways and antiplatelet drug strategies. Initial platelet adhesion to damaged vessel walls is mediated by the binding of collagen to platelet GPVI receptor. Thrombin, generated by the coagulation cascade, is also a potent activator of human platelets through PAR1 und PAR4 receptors. P2Y1 and P2Y12 receptors are stimulated by ADP released from dense granules and the thromboxane prostanoid receptor is stimulated by thromboxane generated by COX-1 signaling pathway. Platelet to platelet aggregation is triggered by fibrinogen and von Willebrand factor binding to αIIbß3 receptor. Direct and indirect (via thrombin inhibition) antiplatelet agents and their targets are shown in the figure above [36,37,38].

**Table 1 ijms-24-06280-t001:** The most important studies on antithrombotic drugs in the last 30 years.

Trial Acronym	Published Year	Short Study Conclusion	Reference
ISAR	1996	Ticlopidine plus aspirin is superior to anticoagulation therapy (heparin or warfarin plus aspirin)	[78]
STAR	1998	Ticlopidine plus aspirin is superior to aspirin alone or aspirin plus warfarin after PCI	[79]
FANTASTIC	1998	Ticlopidine plus aspirin is superior to oral anticoagulation after PCI	[80]
MATTIS	1998	Ticlopidine plus aspirin is safer than oral anticoagulation plus aspirin after PCI	[81]
CLASSIC	2000	Clopidogrel and aspirin is superior to ticlopidine and aspirin	[82]
CURE	2001	Clopidogrel has beneficial effects in NSTEMI	[83]
CREDO	2002	Following PCI, clopidogrel reduces risk of ischemic events	[84]
COMMIT	2005	Adding clopidogrel to ASS improves outcome in ACS	[85]
CLARITY	2005	In STEMI, addition of clopidogrel to standard fibrinolytic therapy and ASS improves outcome	[86]
CHARISMA	2006	Clopidogrel plus aspirin was not significantly more effective than aspirin alone in patients with atherothrombosis	[87]
TRITON	2007	Prasugrel is superior to clopidogrel in ACS	[88]
PLATO	2009	Ticagrelor is superior to clopidogrel in ACS	[89]
CURRENT OASIS 7	2010	7-day double-dose clopidogrel regimen in ACS improved outcome	[90]
REAL-LATE	2010	DAPT longer than 12 months without benefit	[91]
ARCTIC	2012	Bedside platelet function monitoring without influence on outcome	[92]
TRILOGY ACS	2013	Prasugrel is superior to clopidogrel in ACS	[93]
ACCOAST	2013	Prasugrel pre-treatment in NSTEMI increases bleeding risk in CABG	[94]
WOEST	2013	Dual therapy (OAC with clopidogrel) is safer as triple therapy (OAC, ASS and clopidogrel)	[95]
DES-LATE	2014	DAPT with clopidogrel over 12 months does not reduce risk of cardiovascular events	[96]
DAPT	2014	DAPT over 12 months after PCI reduces cardiovascular events but increases bleeding complications	[97]
ISAR-SAFE	2015	No difference in clinical outcome between 6 and 12 months of clopidogrel therapy after DES implantation	[98]
PEGASUS	2015	Treatment with Ticagrelor over 12 months in ACS reduces cardiovascular events	[99]
PRAGUE-18	2016	Prasugrel and Ticagrelor are equivalent in ACS	[100]
NIPPON	2016	Six months of DAPT was not inferior to 18 months of DAPT in DES with a biodegradable abluminal coating	[101]
PIONEER AF	2016	Dual Therapy with Rivaroxaban is safe and causes less bleeding compared to therapy with warfarin	[102]
REDUAL-PCI	2017	Dual Therapy with Dabigatran is safe and causes less bleeding compared to therapy with warfarin	[103]
AUGUSTUS	2019	Dual Therapy with Apixaban is safe and causes less bleeding compared to therapy with warfarin	[104]
ENTRUST AF	2019	Dual Therapy with Edoxaban is safe and causes less bleeding compared to therapy with warfarin	[105]
ISAR-REACT 5	2019	Benefit of Prasugrel in ACS compared to Ticagrelor	[106]

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
