# Peer review of "Novel Aspects Targeting Platelets in Atherosclerotic Cardiovascular Disease—A Translational Perspective"

_ijms, 2023, doi:10.3390/ijms24076280_

Round 1

Reviewer 1 Report

This is a well-written brief review of platelet functions, interactions, pathogenic role, and anti-platelet therapeutic strategies in atherosclerotic cardiovascular pathology. Platelets are central actors in the multicellular pathways of atherothrombosis, and their inhibition has clear prophylactic and therapeutic benefits. The chapters of the manuscript are linked in a logical order, treating important biological processes, some of the complex cellular interactions and, finally, clinical aspects.

Thematically, the review of Huseynov et al. is comprehensive; however, I have several suggestions for its formal and informal improvement.

1. As this work deals exclusively with atherosclerotic cardiovascular disease, I suggest to change the title in: “Novel aspects targeting platelets in atherosclerotic cardiovascular disease – a translational perspective”.

2. Some topics need completion and more focus, with suitable references.

A. The authors describe important steps of endothelial activation, but they do not pay much attention to the vulnerability of atheroma. Plaque rupture denudates the endothelium, can deliver significant quantities of tissue factor and causes the immediate adherence and, further aggregation of platelets. Adhesion and aggregation are different steps of clot formation, involving the platelet GP IIb/IIIa, GP Ib/V/IX and other receptors in a different extent [1,2]. The role of Von Willebrand factor and platelet tissue factor/thromboplastin also worth more attention, maybe with an additional figure included.

B. In the last chapter, the authors mention thrombo-inflammation as a complex pathway, which could be pharmacologically targeted. Low-grade inflammation has a crucial role in atherosclerosis, and may be augmented by thrombocyte PF-4, CD40L and other products. Targeting the activated macrophages and their efflux may improve myocardial contractility in ischemic conditions [Szabo].

C. Dual AP therapy may be helpful, but it can be associated with a thrombelastography-detectable coagulopathy with implications in the management of stroke patients.

With these improvements, I agree with publication of this manuscript in IJMS.

References

 1. Jennings LK. Role of platelets in atherothrombosis. Am J Cardiol. 2009 Feb 2;103(3 Suppl):4A-10A. doi: 10.1016/j.amjcard.2008.11.017

 2. Vorchheimer DA, Becker R. Platelets in atherothrombosis. Mayo Clin Proc. 2006 Jan;81(1):59-68. doi: 10.4065/81.1.59

3. Szabo TM, Frigy A, Nagy EE. Targeting Mediators of Inflammation in Heart Failure: A Short Synthesis of Experimental and Clinical Results. Int J Mol Sci. 2021 Dec 2;22(23):13053. doi: 10.3390/ijms222313053

 4. Hamilos M, Petousis S, Parthenakis F. Interaction between platelets and endothelium: from pathophysiology to new therapeutic options. Cardiovasc Diagn Ther. 2018 Oct;8(5):568-580. doi: 10.21037/cdt.2018.07.01

 5. McDonald MM, Almaghrabi TS, Saenz DM, Cai C, Rahbar MH, Choi HA, Lee K, Grotta JC, Chang TR. Dual Antiplatelet Therapy Is Associated With Coagulopathy Detectable by Thrombelastography in Acute Stroke. J Intensive Care Med. 2020 Jan;35(1):68-73. doi: 10.1177/0885066617729644. Epub 2017 Sep 21. PMID: 28931362.

Reviewer 2 Report

This is an interesting and important review paper that discusses the various aspects/roles of platelets in the context of atherosclerotic disease, be it in terms of therapeutics or preventive measures. The authors try to shed some light on potentially novel therapeutic options that can be employed to inhibit platelets and their interactions that could exacerbate CVD.

This paper is worth publishing but the following minor points need to be addressed before its final acceptance.

·       This reviewer is surprised that endothelial dysfunction and its interplay with platelets in the context of atherosclerosis is not discussed.

·       Perhaps it may be a good idea to add a small paragraph on drugs that are being repurposed for the same goal

·       Adding one more figure would be a plus.

·       If clinical trials can be summed into a table, that may also be helpful.

·       Minor english editing is needed

·       Quality of the cartoons is not optimal; please increase resolution; Figure 2 is a bit crowded and not eye-appealing.

·       Some sentences are without references e.g. lines 97-121
